# The Function of ASK1 in Sepsis and Stress-Induced Disorders

**DOI:** 10.3390/ijms25010213

**Published:** 2023-12-22

**Authors:** John C. Kostyak, Steven E. McKenzie, Ulhas P. Naik

**Affiliations:** Cardeza Center for Hemostasis, Thrombosis, and Vascular Biology, Cardeza Foundation for Hematologic Research, Department of Medicine, Sidney Kimmel Medical College, Thomas Jefferson University, Philadelphia, PA 19107, USA; steven.mckenzie@jefferson.edu (S.E.M.); ulhas.naik@jefferson.edu (U.P.N.)

**Keywords:** sepsis, ASK1, stress, platelet

## Abstract

Apoptosis signal-regulating kinase 1 (ASK1) is a serine-threonine kinase that is ubiquitously expressed in nucleated cells and is responsible for the activation of multiple mitogen-activated protein kinases (MAPK) to regulate cell stress. Activation of ASK1 via cellular stress leads to activation of downstream signaling components, activation of transcription factors, and proinflammatory cytokine production. ASK1 is also expressed in anucleate platelets and is a key player in platelet activation as it is important for signaling. Interestingly, the mechanism of ASK1 activation is cell type-dependent. In this review we will explore how ASK1 regulates a variety of cellular processes from innate immune function to thrombosis and hemostasis. We will discuss how ASK1 influences FcγRIIA-mediated platelet reactivity and how that reactivity drives platelet clearance. Furthermore, we will explore the role of ASK1 in thromboxane (TxA_2_) generation, which highlights differences in the way ASK1 functions in mouse and human platelets.

## 1. Introduction

ASK1 belongs to a group of upstream kinases of the MAPK signaling cascade, known as MAP3Ks, that are responsible for phosphorylating MAP2Ks. MAP2Ks in turn, phosphorylate terminal MAPKs like extracellular signal-regulated kinase (ERK1/2), c-Jun N-terminal kinase (JNK1/2), and p38 MAPK. MAPKs are highly conserved serine/threonine kinases, which are expressed in nucleated cells and platelets [1,2,3]. In nucleated cells, MAPK downstream signaling regulates cell survival, differentiation, and cell proliferation. ERK1/2 is activated by growth factor stimulation and regulates proliferation, while p38 and JNK respond to cell stress and regulate cell survival and inflammation. This signaling culminates in the activation of transcription factors and the production of proteins that dictate cell behavior. The activation of MAP kinase kinase kinase kinase (MAPK4K, MEKKK) or MAP kinase kinase kinase (MAP3K, MEKK) occurs via phosphorylation or association with small GTPases. MAP4Ks and MAP3Ks then activate MAP2Ks and MAP2Ks activate MAPK through the phosphorylation of two residues (threonine and tyrosine) that are separated by one amino acid (Thr-X-Tyr) [4].

ASK1 appears to control several aspects of cellular signaling as it has over 90 potential protein interacting partners [5]. ASK1 is a key player in cellular stress response to a multitude of stimuli in a variety of cell types [6,7,8,9]. Perhaps the best-characterized role for ASK1 involves toll-like receptor (TLR) engagement of a pathogen-associated molecular pattern (PAMP), reactive oxygen species (ROS) generation, and activation of downstream transcription factors that lead to proinflammatory cytokine production. Blocking ASK1 function by genetic ablation, pharmacological inhibition, or inhibition of upstream factors reduces p38 phosphorylation and TLR-mediated cytokine production, establishing ASK1 as a critical signaling component of innate immunity [9,10,11]. ASK1 hyperactivity is linked with nonalcoholic steatohepatitis (NASH), the more severe form of nonalcoholic fatty liver disease, which is characterized by ROS generation and endoplasmic reticulum (ER) stress [12,13]. Furthermore, it was recently demonstrated that ASK1-p38 signaling is required for the replication of several different viruses (which generate ROS in infected cells) in human cell lines, as ASK1 inhibition with either GS444217 or Selonsertib disrupted viral replication [14]. The above examples highlight the ASK1-p38 signaling axis and its importance to the cell stress response. Recent reviews summarize the role of ASK1 in nucleated cells [15,16]. We demonstrated that Ask1 is expressed in platelets even though platelets lack a nucleus and ASK1 regulates platelet signaling in response to a variety of physiological agonists [17,18,19]. Below, we discuss this important signaling mechanism and how it relates to the cell stress response and platelet activation.

## 2. ASK1

ASK1, also known as mitogen-activated protein 3 kinase 5 (MAP3K5), was identified by Ichijo et al. using a degenerate primer PCR-based strategy [20]. We then showed that ASK1 is expressed in both human and murine platelets and that ASK1 regulates platelet reactivity [17,18]. In platelets, MAPKs are activated through various surface receptors following the engagement of their physiological ligands and are essential for platelet activation and subsequent thrombus formation. If the MAPK signaling cascade that activates ASK1 in nucleated cells were to hold true in platelets, ASK1 would be activated by MAP4Ks and would phosphorylate MAP2Ks (MKK4 and MKK3/6), which would activate p38 [18]. However, the only requirement for ASK1 activation in platelets (and dopaminergic neurons) is Ca^2+^ rise, suggesting a distinct dichotomy in ASK1 activation depending on cell type [19]. We will explore that mechanism in this review. 

## 3. ASK1 Structure

Human ASK1 is a protein containing 1374 amino acids that is composed of three domains. A thioredoxin binding domain is located near the N-terminus and is, as the name implies, the binding site for thioredoxin [21,22]. The central regulatory region contains the tumor necrosis factor receptor-associated factor (TRAF) binding domain and the catalytic domain, while a coiled-coil domain can be found near the C-terminus as well as the N-terminus [23,24]. Crystallographic information for ASK1 is not entirely available, though some regions have been revealed. The central regulatory region contains a pleckstrin homology domain and 14 helices which form seven tetratricopeptide repeats [23]. Furthermore, the catalytic domain contains a small lobe with five β sheets, an α helix, and a larger C-terminal lobe made of mostly α helices [6]. ASK1 also contains a 14-3-3 binding motif near the C-terminal end [6]. 

## 4. ASK1 Activation 

### 4.1. In Most Nucleated Cells

ASK1 is ubiquitously expressed and is activated in a cell type-dependent manner [20]. In most nucleated cells (neurons being the exception) ASK1 mediates apoptosis and is activated by the oxidation of thioredoxin via ROS [20,21,25,26,27]. During non-stress conditions, ASK1 binds 3 dithiol reductases that respond to redox stimuli [28,29,30]. Thioredoxin and peroxiredoxin bind towards the N-terminus of ASK1, while glutaredoxin binds towards the C-terminus [21,31,32]. These dithiol reductases along with 14-3-3 bind to ASK1 and form a multiprotein complex. In the absence of stress stimuli interaction of ASK1 with thioredoxin promotes ASK1 Lys48-linked polyubiquitination and proteolytic degradation [33,34]. However, redox reactions induced by ROS formation result in disulfide bond formation that releases each dithiol oxidoreductase from ASK1, which allows subsequent autophosphorylation and activation of ASK1 [35,36]. Interactions with TRAF6 promote non-proteolytic ubiquitination of ASK1, which accelerates the dissociation of thioredoxin and promotes ASK1 N-terminal dimerization resulting in enhanced activation of the JNK and p38 signaling cascades in hepatocytes [37]. Consistently, it was reported that ASK1 interacts with the deubiquitinase tumor necrosis factor alpha-induced protein 3 (TNFAIP3), which deubiquitinates ASK1 to suppress its activity [38]. Once activated ASK1 promotes apoptosis via signaling, which results in the activation of p38 protein kinases (p38) and JNK1/2 [20]. Interestingly, ASK1 regulation is quite different in platelets and neurons, where it is constitutively bound to calcium- and integrin-binding protein 1 (CIB1) [19]. 

### 4.2. In Platelets

Platelets, produced by megakaryocytes, are anucleate cells that mediate hemostasis by amplifying an initial stimulus and aggregating at a site of injury. Platelets are involved in many processes ranging from fighting microbial infections and triggering inflammation to promoting tumor angiogenesis and metastasis [39,40,41,42,43,44,45]. Nevertheless, the primary physiological function of platelets is to act as an essential mediator in maintaining hemostasis by forming hemostatic thrombi that prevent blood loss and maintain vascular integrity [46,47]. When there is vascular damage, exposure of the extracellular matrix recruits and activates platelets thereby leading to aggregation and formation of a fibrin-rich hemostatic plug at the injured site. Platelet aggregation is achieved by fibrinogen-dependent adhesion through activation of α_IIb_β_3_ on the platelet surface. 

While the classical function of platelets is to prevent blood loss (hemostasis), the role of platelets in immunity has begun to take shape. Platelets arrive quickly at the sight of infection and are key recruiters of other immune cells [48,49]. Platelets migrate to and bundle bacteria, which enhances the phagocytic activity of neutrophils [50]. Furthermore, platelets can kill bacteria via phagocytosis and secretion of antibacterial peptides that come from alpha granules [51,52]. The bactericidal potential of platelets is augmented by its activation in vitro [53]. 

Platelets are produced by megakaryocytes in a process that resembles apoptosis. As megakaryocytes mature ASK1 expression increases along with other pro-apoptotic proteins suggesting that ASK1 is important for thrombogenesis [54]. In resting platelets, ASK1 is bound to CIB1. Agonist stimulation of platelets that results in Ca^2+^ release triggers the dissociation of CIB1 from ASK1 and subsequent ASK1 autophosphorylation [19]. Previously, we determined that ASK1 is activated in platelets by Ca^2+^ release regardless of the agonist used to stimulate the platelets [19]. In fact, agonists that stimulate both G-protein-coupled receptors (GPCR) and immunoreceptor tyrosine-based activation motif (ITAM) receptors elicit ASK1 phosphorylation. Although there is an abundance of thioredoxin in platelets it is not bound to ASK1 as it is in most nucleated cells suggesting that ROS may not be sufficient to cause ASK1 phosphorylation in platelets. Instead, we showed that modification of cytosolic Ca^2+^ using thapsigargin, which stimulates Ca^2+^ release from stores caused ASK1 phosphoylation, while chelating Ca^2+^ using BAPTA-AM reduced ASK1 phosphorylation in platelets. Thus, we concluded that ASK1 is phosphorylated and activated due to a rise in intracellular Ca^2+^ [19]. This association of ASK1 and CIB1 is disrupted as intracellular Ca^2+^ concentrations rise due to agonist stimulation. As ASK1 and CIB1 dissociate, ASK1 then associates with TRAF6 and becomes autophosphorylated, suggesting that CIB1 binds and inhibits ASK1 until Ca^2+^ is available to bind CIB1, which liberates ASK1 [19]. In confirmatory studies, *Cib1^−/−^* mouse platelets have enhanced p38 phosphorylation while resting and following agonist stimulation because Cib1 is not there to sequester ASK1 [19]. 

### 4.3. In Neurons

Just as in platelets, ASK1 interacts with CIB1 in dopaminergic neurons and CIB1 serves to inhibit the binding of TRAF2 to ASK1 [55]. Using dopaminergic neuroblastoma cells (SH-SY5Y), we showed that ASK1 mediates cell death due to neurotoxin 6-hydroxydopamine (6-OHDA) and that knockdown of CIB1 exacerbates the apoptotic response [26]. However, pretreatment with BAPTA-AM, a Ca^2+^ chelator, in CIB1 expressing SH-SY5Y cells treated with 6-OHDA and KCl was protective, suggesting that intracellular Ca^2+^ rise caused CIB1 to dissociate from ASK1, allowing ASK1 activation. Similar results were obtained using primary rat mesencephalic dopaminergic neurons [26]. These data suggest that ASK1 is regulated similarly in neurons and platelets. 

## 5. ASK1 Function

### 5.1. In Stress

As the name implies, ASK1 is involved in the apoptotic process by inducing stress-mediated activation of JNK1/2 and p38 [20]. Tumor necrosis factor-α (TNF-α) exposure was sufficient to activate ASK1 in several stable cell lines expressing ASK1 [20]. Furthermore, overexpression of ASK1 exacerbated stress-induced cell death, while a kinase-dead mutant acted as a dominant negative [20]. These data suggest that ASK1 is involved in apoptosis and is activated by proinflammatory cytokines. It was later shown that ASK1 is also involved in genotoxic cell death as the DNA-damaging agent cisplatin activated ASK1 and downstream p38 leading to caspase activation and apoptosis of human ovarian carcinoma and kidney cells [56]. Overexpression of kinase-negative (K709R) ASK1 decreased apoptosis in these cell lines [56]. Furthermore, it appears as though the action of ASK1 in stress-induced apoptosis involves the mitochondria as constitutively active ASK1 induces apoptosis via caspase activity and cytochrome C release [57]. Caspase 9 is involved as murine embryonic fibroblasts lacking caspase 9 are resistant to ASK1-induced apoptosis [57]. 

MAPKs are part of the host cell stress response to viral infection. Specifically, p38 MAPK phosphorylation occurs following infection of severe acute respiratory syndrome coronavirus 2 (SARS-CoV-2), Human Immunodeficiency Virus (HIV), and Herpes Simplex Virus (HSV) [58,59,60]. Using the vaccinia virus (VV), HSV, and the RNA vesicular stomatitis virus (VSV), Demian et al. showed that p38 phosphorylation was induced in two infected human cell lines [14]. Inhibition of ASK1 using selonsertib had a dramatic negative effect on viral replication of HIV, HSV, VSV, and SARS-CoV-2 [14]. However, this did not translate to a hamster model as selonsertib enhanced SARS-CoV-2 replication [14]. It would be interesting to determine how reduction in viral replication via ASK1 inhibition is species-specific. 

The MAPK signaling cascade involving ASK1/JNK/p38 is also implicated in a variety of neurodegenerative diseases. Using an established mouse model (SOD1^G93A^) of amyotrophic lateral sclerosis (ALS) Fujisawa et al. reported that inhibition of ASK1 extended the lifespan of these mice [61]. ASK1 was identified as a modifier gene for the age of onset of Huntington’s disease (HD) and inhibition of ASK1 in a mouse model of HD led to reduced atrophy and motor dysfunction [62,63]. Furthermore, inhibition of the ASK1 signaling pathway is reported to improve several aspects of Parkinson’s disease [64,65,66]. Finally, ASK1 mediates neuronal cell death via amyloid-β, which leads to cognitive impairment in Alzheimer’s disease [67,68]. 

### 5.2. In Sepsis

In the United States, 33% of patients who die in a hospital have sepsis and worldwide the mortality rate for sepsis patients is 26% [69]. Sepsis is listed as the single most expensive disease in hospitals within the United States and is characterized as the body’s overwhelming response to infection [69]. As sepsis advances it can lead to septic shock, which is a hemodynamic disorder characterized by very low blood pressure and is concomitant with thrombocytopenia caused, among other reasons, by the formation of immune complexes (IC). Mortality due to septic shock is high [70]. 

When local infections are not contained and progress to systemic infections, a rapid inflammatory response occurs. Sepsis occurs when PAMPs such as lipopolysaccharide (LPS) from gram-negative bacteria or peptidoglycan from gram-positive bacteria interact with pattern recognition receptors (PRRs) such as TLRs on inflammatory cells [71]. This causes the release of pro-inflammatory and anti-inflammatory cytokines that define the dysregulated host response to infection. Platelets also express PRRs like TLR4 that respond to PAMPs. Using platelet TLR4 as an example, it responds to LPS and elicits a variety of responses from the production of TNF-α to enhanced interaction with neutrophils. This combined with inflammatory cell response results in a feed-forward mechanism that causes super physiological levels of circulating cytokines and triggers the ongoing activation of platelets and inflammatory cells like neutrophils and monocytes [72]. 

Key components of the dysregulated host response that defines sepsis are TLRs, which are expressed on various immune cells and platelets. Stimulation of TLRs leads to recruitment or stimulation of MyD88, TRAF, and TIR domain-containing adaptor protein (TIRAP) followed by formation and phosphorylation of the transforming growth factor-β-activated kinase (TAK1)/TAK1 binding protein (TAB) complex [73]. This leads to signaling culminating in JNK1/2, p38, and nuclear factor-κB (NF-κB) activation. The TLR signaling pathway is commonly interrogated using LPS. Using the antioxidants AGI-067 and AGI-1095 Luyendyk et al. demonstrated that inhibition of ROS production following LPS stimulation in monocytic and endothelial cells reduced ASK1 activity but did not impact nuclear translocation of NF-κB [7]. The MAPKs p38, JNK, and ERK1/2 were all inhibited. Also, using LPS, Matsuzawa et al. demonstrated that ASK1 is required for p38 activity in the RAW 264.7 macrophage cell line [10]. Furthermore, the requirement for ASK1 extended to the production of proinflammatory cytokines, as deletion of Ask1 in mouse splenocytes significantly reduced TNF-α, IL-6, and IL-1β production [10]. This translated into the protection of *Ask1^−/−^* mice from LPS-induced mortality. Similarly, it was reported that inhibition of ROS production resulting from LPS stimulation of RAW264.7 macrophages prevented the formation of the TRAF6-ASK1 complex and subsequent p38 activation and production of proinflammatory cytokines [11]. Miller et al. demonstrated that inhibition of ASK1 in endothelial cells significantly reduced LPS-stimulated cytokine production without any effect on endothelial permeability [9]. Recently, it became clear that platelets play more than just a passive role in immunity as they can recognize and kill bacteria [50]. Furthermore, platelets are important as thrombocytopenia, which is associated with thrombotic shock, results in enhanced mortality in critically ill septic patients [74]. It will be interesting to determine whether ASK1 has any role in platelets during sepsis. Several bacterial species are known to activate platelets through the FcγRIIA receptor leading to aggregation, secretion, and spreading [75,76]. The FcγRIIA receptor recognizes IgG-opsonized bacteria and engagement of IgG with FcγRIIA causes rapid phosphorylation of the ITAM and subsequent downstream signaling. 

### 5.3. In Hemostasis and Thrombosis

Deletion of Ask1 in mice results in impaired platelet activation, interrupted hemostasis, and protection from thrombosis [17]. In mouse platelets, Ask1 is activated following stimulation by physiological agonists and when Ask1 is ablated, platelets display reduced aggregation to physiological agonists characterized by attenuated α_IIb_β_3_ activation [17]. This is independent of Ca^2+^ as there are no alterations in Ca^2+^ rise in *Ask1^−/−^* mouse platelets following agonist stimulation [17]. However, granule secretion, as measured by C^14^-serotonin release from dense granules and P-selectin exposure from α-granules, and thromboxane (TxA_2_) production were significantly reduced in *Ask1^−/−^* platelets following agonist stimulation [17]. A likely explanation behind reduced TxA_2_ generation and impaired platelet reactivity is the fact that p38 MAPK is not phosphorylated in an agonist-dependent manner in *Ask1^−/−^* platelets, while phosphorylation of other MAPKs was enhanced [17]. This suggests that in mouse platelets Ask1 is required for p38 phosphorylation but not JNK1/2 or ERK1/2 phosphorylation. Homozygous genetic ablation of p38 in mice results in death in utero, while heterozygous mice have impaired thrombosis [77,78]. This suggests that ASK1 and p38 are part of the same signaling pathway that is separate from other MAPKs in platelets. Lack of p38 activity results in reduced cytosolic phospholipase-A_2_ (cPLA_2_) phosphorylation and impaired TxA_2_ generation [79]. As such, *Ask1^−/−^* mice experience interrupted hemostasis and are protected from thrombosis [17]. Tail bleeding times were enhanced in *Ask1^−/−^* mice, suggestive of impaired hemostasis, while carotid artery occlusion time following FeCl_3_-induced injury was increased and survival following collagen/epinephrine-induced pulmonary thromboembolism was extended in *Ask1^−/−^* mice, suggestive of protection from thrombosis [17]. 

### 5.4. In Heparin-Induced Thrombocytopenia

ICs form when an antibody is bound to an antigen and are present during sepsis, heparin-induced thrombocytopenia (HIT), and lupus erythematosus [75,80,81,82]. Like sepsis, patients who develop HIT have a high (10%) mortality rate [83]. Therefore, there is a great need for increased understanding of IC actions. In terms of HIT, ICs can bind to FcγRIIA, which is expressed on platelets (and other cells), inducing platelet activity leading to immune-induced thrombocytopenia and thrombosis [18]. When ICs bind, FcγRIIa is phosphorylated by Src family kinases (SFKs) on the tyrosine residue of its ITAM upon activation [84,85,86]. Syk then binds to the phosphorylated ITAM through its two SH2 domains and becomes autophosphorylated. Tyrosine phosphorylation of Syk leads to phosphorylation of several adaptor proteins including phospholipase C γ2 (PLCγ2), linker for T-cell activation (LAT) and Src homology 2-containing leukocyte protein 76 (SLP76), activation of phosphoinositide 3-kinase (PI3 kinase), and recruitment of Bruton’s tyrosine kinase (Btk). Recent studies by others have also shown that PI3 kinase β has an important role in GPVI-mediated platelet activation [87,88]. The signaling initiated by FcγRIIA activation results in Ca^2+^ influx, TxA_2_ generation, secretion of granule contents, and activation of α_IIb_β_3_. The exact consequence of IC-mediated platelet activation as it relates to conditions like HIT is under investigation. To study HIT in a more comprehensive fashion, humanized mice expressing FcγRIIA were produced, as mice do not normally express this protein [89]. We used these mice to study the interplay between ASK1 and FcγRIIA.

It is known that ASK1 positively regulates FcγRIIA-mediated activation of human and humanized mouse platelets [18]. By crossing humanized *FcγRIIA* transgenic mice (*hFcR/Ask1^+/+^*) with *Ask1^−/−^* mice we produced *hFcR/Ask1^−/−^* mice. Using this line, we demonstrated that Ask1 is activated in a FcγRIIA-dependent manner as characterized by Ask1 phosphorylation and phosphorylation of downstream targets [18]. Inhibition of ASK1 in human platelets impaired FcγRIIA-mediated aggregation suggesting that ASK1 is integral in the platelet response to immune complexes [18]. Similar results were obtained in murine platelets from *hFcR/Ask1^−/−^* mice. Signaling proximal to the receptor was intact as PLCγ2 and Syk were both phosphorylated following mouse anti-CD9 stimulation, but downstream signaling involving p38 and cPLA_2_ was markedly reduced in *hFcR/Ask1^−/−^* platelets compared to control, which explains reduced TxA_2_ generation in this model. Therefore, Ask1 regulates FcγRIIA-mediated platelet reactivity by enhancing TxA_2_ generation (Figure 1). Consistently, we showed that Ask1 is vital for immune complex-induced shock, thromboembolism, and thrombocytopenia [18]. For the remainder of this review, we will focus on aspects of our previous work that raised new questions. 

## 6. ASK1 and Platelet Clearance

Platelet clearance in the liver or spleen and platelet production via megakaryocytes exist in a balance to prevent occlusive thrombi and spontaneous bleeding. Platelets are the second most abundant cell in circulation behind only red blood cells. As such, billions of platelets are produced and cleared physiologically each day. However, activation of the IC, as occurs in HIT, vaccine-induced immune thrombotic thrombocytopenia, and sepsis, disrupts this balance by leading to platelet activation that results in thrombi formation, disseminated intravascular coagulation, and subsequent thrombocytopenia [90,91,92]. 

We recently demonstrated that platelet clearance following stimulation via FcγRIIA is tied to platelet activation, which is regulated by ASK1 [18]. For instance, when low doses of mouse anti-CD9 are used (460–490 ng/mL) to stimulate isolated platelets we found that Ask1 deletion caused reduced aggregation compared to *hFcR/Ask1^+/+^* mouse platelets. When increased concentrations (>500 ng/mL) of mouse anti-CD9 are used in isolated platelets, there is little to no difference in platelet reactivity in *hFcR/Ask1^−/−^* platelets compared to *hFcR/Ask1^+/+^*. Similarly, platelet clearance is reduced in *hFcR/Ask1^−/−^* mice at a low mouse anti-CD9 concentration (2.5 μg/mouse) administered in vivo. But, when higher concentrations (10 μg/mouse) of mouse anti-CD9 are administered in vivo there is no difference in platelet clearance. These data strongly suggest that platelet clearance is tied to platelet activity in *hFcR/Ask1^−/−^* mice. The fact that platelet clearance seems to correlate with platelet reactivity raises the potential of an ASK1 inhibitor as a therapeutic.

## 7. The Involvement of ASK1 in ITAM-Mediated Signaling

We recently demonstrated that ASK1 is rapidly phosphorylated when human platelets are stimulated with either CD9 antibody or anti-hFcγRIIA (IV.3) + goat anti-mouse (GAM) [18]. This follows the known mechanism by which ASK1 is activated, which is via Ca^2+^-dependent CIB1 dissociation and subsequent autophosphorylation [19]. Of note is the rapid dephosphorylation of ASK1 following peak phosphorylation approximately 1 min after stimulation, particularly with IV.3 + GAM [18]. Clearly, there is tight control over ASK1 phosphorylation during platelet activation through FcγRIIA. Signaling cascades are a delicate balance of protein activation and inhibition which often occurs through phosphorylation and dephosphorylation. A closer examination of a potential ASK1 phosphatase during platelet activation in this scenario could prove useful. Two potential, well-known platelet phosphatases that interact with ASK1 are protein phosphatase-2A PP2A and Src homology region 2 domain-containing phosphatase-1 (SHP-1) [93,94]. Both phosphatases have not only been established in platelets but also function within the same signaling pathway as ASK1 [95,96,97]. It would be very interesting to determine if one of these phosphatases is responsible for the rapid dephosphorylation of ASK1 observed following ITAM stimulation. As ASK1 regulates TxA_2_ generation, hyperphosphorylation of ASK1 could lead to continuous generation of TxA_2_, greater than necessary platelet activation, and pathological thrombus formation. Therefore, tight control over ASK1 phosphorylation via phosphatase activity may be crucial to allow hemostasis but restrict thrombosis. 

## 8. ASK1 and TxA_2_ Generation in Human Platelets

Ask1 regulates TxA_2_ generation and serotonin secretion in mouse platelets as well as humanized mouse platelets activated via FcγRIIA [17,18]. In a recent report from Sledz et al., they demonstrated that TxA_2_ generation in human platelets was not altered by ASK1 inhibition [98]. However, the authors used a single high dose of collagen-related peptide (CRP-XL) (5 μg/mL) for both their human and murine studies. We have previously shown that the effects of ASK1 inhibition or deletion can be overcome when high concentrations of agonists are used to stimulate either human or murine platelets [17,18]. It is possible that TxA_2_ generation from human platelets pretreated with an ASK1 inhibitor will be reduced compared to control platelets when stimulated with lower concentrations of agonists. Evidence for this exists in the report by Sledz et al. [98]. They report reduced TxA_2_ generation in mouse platelets stimulated with CRP-XL following pretreatment with the Ask1 inhibitor GS4997 compared to controls, but no difference in human platelets pretreated with GS4997 compared to controls. However, there was at least five times more TxA_2_ generated by the human platelet controls compared to the mouse platelet controls. Therefore, a CRP-XL dose–response curve may have revealed a role for ASK1 in TxA_2_ generation in human platelets. This would be particularly interesting given that TxA_2_ generation has long been implicated in sepsis and subsequent septic shock [99]. 

We recently reported that humanized mice lacking Ask1 were protected from thrombotic shock [18]. This is likely due to reduced TxA_2_ generation and subsequent reduced granule secretion that results in decreased platelet reactivity following stimulation via FcγRIIA. TxA_2_ generation is important for sepsis progression as well as FcγRIIA-mediated platelet activation [18,99]. Consistently, we previously reported a critical role for FcγRIIA in TxA_2_ generation, which occurs via calcium- and diacylglycerol-regulated guanine nucleotide exchange factor-1 (CalDAG-GEF1) [100]. As mentioned above, ASK1 regulates platelet secretion via TxA_2_ generation, which is the likely explanation for the observed reduced thrombotic shock in *hFcR/Ask1^−/−^* mice [18]. 

## 9. Conclusions

The data discussed above point to ASK1 having a prominent role in multiple aspects of cellular function. ASK1 is regulated differently in anucleate platelets and dopaminergic neurons compared to other nucleated cells in that it binds CIB1, which responds to Ca^2+^ rise in the former and binds thioredoxin, which responds to ROS in the latter. Additionally, ASK1 regulates the cellular response to PAMPS in multiple cell types, making it an excellent therapeutic target for sepsis. Similarly, there is the potential for ASK1 as a therapeutic target for the treatment of several neurodegenerative disorders. Also, ASK1 and its downstream signaling targets are involved in viral replication. ASK1 regulates FcγRIIA-mediated platelet activation, which drives platelet clearance during HIT, suggesting that ASK1 may be a relevant target for HIT. Furthermore, ASK1 regulates ITAM-mediated signaling and is rapidly dephosphorylated shortly thereafter by an as-yet-unknown phosphatase. Finally, ASK1 regulates murine TxA_2_ generation, and we believe that a more comprehensive investigation into the role of human platelet TxA^2^ generation is warranted. It is clear that ASK1 is a prominent cell signaling protein that represents a potential therapeutic target for multiple disease states. 

## Figures and Tables

**Figure 1 ijms-25-00213-f001:**
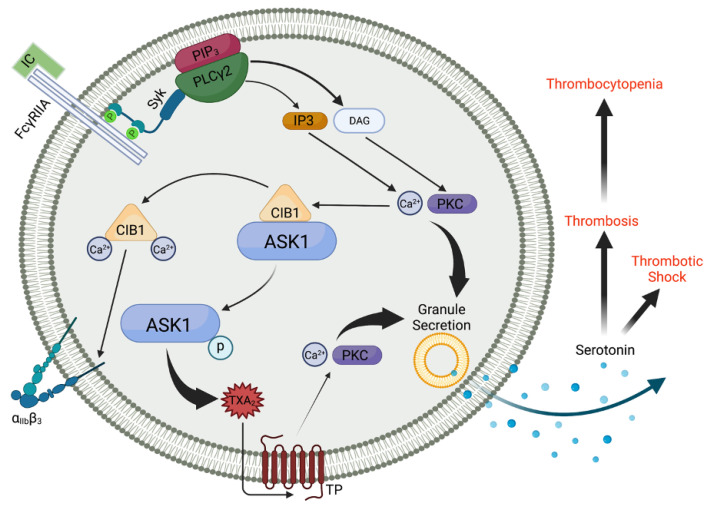
ASK1 regulates thromboxane generation and subsequent granule secretion downstream of FcγRIIA in human platelets. FcγRIIA receptor engagement leads to phosphorylation of tyrosine residues within the dual YXXL motifs found on ITAMs. This leads to Syk recruitment and a subsequent rise in intracellular Ca^2+^, which liberates ASK1 from CIB1, resulting in ASK1 autophosphorylation, which is required for its activation. While CIB1 binds and causes α_IIb_β_3_ activation, ASK1 regulates TxA_2_ generation via the p38/cPLA2 pathway. TxA_2_ signals through the thromboxane prostanoid receptor (TP) and is a key mediator of platelet granule secretion. Serotonin, a component of dense granule contents, is known to enhance septic shock due to increased endothelial permeability and vasodilation. Serotonin and ADP, released from dense granules, enhance endothelial dysfunction and platelet reactivity, respectively. Created with BioRender.com.

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
