# Peer review of "The Function of ASK1 in Sepsis and Stress-Induced Disorders"

_ijms, 2023, doi:10.3390/ijms25010213_

Round 1

Reviewer 1 Report

Comments and Suggestions for Authors

The authors reviewed the role of ASK1 in different pathologies, especially in platelet response. This review is well-prepared and written, although it can be improved in some respects.

Major points:

-          The weakest point of the review is that it refers to the differences between humans and mice, which should be more developed and better explained. As I understand There are two main differences: mice lack FcRIIA, but ASK1 regulates TxA2 generation, whereas in humans, it is not clear if ASK1 regulates TxA2. Then, What animal species is the graph in Figure 1 for, would it apply to humans?

-          I think there should be a section that includes neurodegenerative diseases in section 5. Neurons have ASK1 activation that is different from that of other cells and is similar to that of platelets. ASK1 may be involved in ALS, HD, Parkinson's disease, and even Alzheimer's disease. Their involvement in these pathologies can be further explained.

Minor points:

-          Line 92, the ROS acronym has already appeared before in the text.

-          Please revise the statement on lines 101–103. If TRAF6 promotes ubiquination, ASK1 would be degraded and not activate JNK and p38, right?

-          Line 170, please replace “kinase dead” for “kinase-negative (K709R) ASK1”

-          I would recommend removing the phrase from lines 206 to 209. Please note that LPS activation during sepsis triggers a complex response with a mix of inflammatory and anti-inflammatory cytokine production (including IL-10) via mechanisms such as endotoxin tolerance (Immunity. 2015;42(3):484-498. doi:10.1016/j.immuni.2015.02.001).

-          Line 268. Please do not use an acronym in this title.

-          Revise FcγRIIA has the A in capital and ASK1 as well is in capital along text (line 275 and line 350 respectively)

-          Line 342 “these phosphatases is are responsible

Comments on the Quality of English Language

Quiality of English is fine.

Reviewer 2 Report

Comments and Suggestions for Authors

The review of Kostyak and coauthors is devoted to the discussion of the role of ASP1 MAP kinase in the development of sepsis. The review is well-written and well-structured and can be accepted for publication in the IJMS after minor corrections. However, it seems to me that adding/changing the following points may make it easier for the readers to understand the material.

Minor points.

1.       Authors mention ASK1-signalosome in the text, however, its structure was not explicitly described. Keeping in mind that the term “ASK1-signalosome” is ambiguous (the authors name the inhibitory complex ASK1-thioredoxin-14-3-3 “signalosome”, while other authors name the active ASK-1-TRAF6-MyDD88 “signalosome”, for ex. see https://doi.org/10.1074/jbc.M506771200) , it is advisable to describe it in more details.

2.       In line 33 it is stated that ubiquitination leads to ASK1 degradation, while in line 101 it is stated that ASK1 ubiquitination is needed for its activity, and in line 105 – that ASK1 deubiquitination leads to its inactivation. The effects of ASK1 ubiquitination should be described in more details, at least, some opinion on its role should be given. Especially keeping in mind that ubiquitine ligases are very active in platelets.

3.       The authors state that in platelets ASK1 is complexed with a protein called CIB1, and some data are given that no thioredoxin-ASK1 complexes were found in platelets. This information is not enough, as ASK1 contains 14-3-3-binding domain and thioredoxin-binding domain (ASK1-TBD), and both 14-3-3 and thioredoxin binding block ASK1 activity (https://doi.org/10.3390/ijms222413395). Therefore, CIB1 binding site and its effects on ASK1 activation should be discussed.

4.       Figure 1 implies that ASK1 is activated in platelets upon activation of FcRIIA. However, murine platelets do not express Fc receptors (except for humanized mice). But many conclusions in the review are drawn from wild type murine models. On the other hand, ASK1 activation from either TLR4 or IL-1R is well established, and platelets contain both of these receptors. These discrepancies should be clarified before the review could be published.

5.       On the other hand, the authors argue that ASK1 in platelets is activated through calcium signaling. However, than it could be activated by all other platelet activating agents (thrombin, collagen, etc.), as all of them induce calcium signaling. But in this case inhibition of ASK1 will influence all platelet activation responses, not only those involved in sepsis.

6.       Some sentences in the conclusions should be corrected:

a.       “ASK1 is regulated differently in anucleate platelets and dopaminergic neurons compared to other nucleated cells.” – the authors did show that CIB1 regulate ASK1 in platelets and neurons, but whether it regulates ASK1 in other cells is not clear.

b.      “Furthermore, ASK1 regulates ITAM-mediated signaling and is rapidly dephosphorylated shortly thereafter by an as-yet unknown phosphatase.” – the authors did not show that ASK1 regulates GPVI-mediated signaling in platelets or other ITAM-containing receptors in other cells, therefore, the statement is premature. Additionally, in other cells ASK1 is known to be regulated by DUSP3 and other phosphatases, which are present in platelets; therefore, some discussion on the identity of phosphatase, should be given.  
